# OpenReview forum: "Prompt Injection Benchmark for Foundation Model Integrated Systems"
_ICLR.cc/2025/Conference — Submitted to ICLR 2025_

### Official Review · Reviewer_rLXR · 2024-10-30

**Soundness:** 3
**Presentation:** 3
**Contribution:** 2
**Rating:** 5
**Confidence:** 2

**Summary:**

This paper present a new framework for benchmarking systems that integrate a foundation model (FM). The framework considers both FM-integrated applications and FM-integrated agents; it spans multiple modalities, including code and vision, and multiple tasks, including web- and code-agent tasks. Attacks are evaluated in terms of information- and task-level threats, such as information leakage and adversarial actions. The paper uses the framework to evaluate three models (GPT-4o mini, Llama 3 as LLM, and Qwen-VL as VLM) against a number of prompt-injection variants (e.g., direct attack, boundary-confusion attack) and defenses (e.g., sandwich prevention, data isolation), showing that prompt-injection attacks may still pose a significant threat.

**Strengths:**

* More comprehensive benchmark than existing ones in terms of modality, types of threats, and evaluation pipeline.
* Benchmarking results on some state-of-the-art models as well as on baseline attacks and defenses.
* Prompt injection is an important problem.

**Weaknesses:**

* The implementation of the framework does not seem to be available.
* The evaluation of usability is not rigorous, i.e., paper claims that the framework is "user-friendly" but this is not evaluated (e.g., using user study).
* It would be better if the attacks and defenses listed in Section 3.3.2 were accompanied by references.

**Questions:**

* Would it make sense to frame the three information-level threats (leakage, goal hijacking, and refusal) in terms of the traditional CIA (confidentiality, integrity, and availability) triad? Also, is the terminology of "threat levels" common in the literature? The usage of the word "threat" in Section 3.2 seems unconventional.
* "despite the claims of strong safety alignment in the Llama 3 model, it still exhibits vulnerabilities to prompt injection"
Is safety alignment supposed to prevent prompt injection?

---

### Official Review · Reviewer_CK4W · 2024-10-31

**Soundness:** 3
**Presentation:** 3
**Contribution:** 3
**Rating:** 6
**Confidence:** 4

**Summary:**

This paper introduces FSPIB, a comprehensive benchmark for evaluating prompt injection vulnerabilities in FM-integrated systems. FSPIB addresses gaps in current benchmarks by covering diverse modalities beyond text, including code, web, and vision, and by providing a dynamic, interactive testing environment. It evaluates attacks and defenses in agent-based systems, enabling ongoing adaptation to evolving adversarial strategies. Through baseline analysis, FSPIB demonstrates the prevalence of security vulnerabilities in FM-integrated systems and the limited efficacy of current defenses, highlighting the need for further research in prompt injection mitigation.

**Strengths:**

Strengths:
+ This work offers FSPIB, a comprehensive benchmark for evaluating prompt injection vulnerabilities in FM-integrated systems.
+ Comprehensive and thorough assessment.
+ Revealed the prompt injection security risk of the LLM agent framework.

**Weaknesses:**

Weaknesses:

- Lack of evaluation on real-world FM systems as mentioned in the limitations of the manuscript.
- Lack of design of attacks or defenses.
- The information provided in the manuscript is difficult to reproduce the proposed evaluation benchmark.

**Questions:**

Questions:

- Q1: Regarding the evaluation metrics, the attack success rate defined in this paper seems a bit vague. It would be better if the authors could provide more practical examples.

- Q2: As for the examples of successful attacks, the article seems to lack the demonstration of these cases. It would be better if the authors could show more examples of attacks.

- Q3: The concepts of "application" and "agent" in this article require further clarification. For instance, in the context of a web application, it is unclear whether the application performs a series of web-related operations or is solely intended for web retrieval. To enhance reader understanding of the practical implications of the attack scenarios and the interpretation of attack success rates, the authors should provide a more detailed explanation of the application scenarios and purposes for each application in the appendix.

- Q4: In addition to the baseline attacks and defenses, have the authors considered proposing any new attack and defense mechanisms? Given that ICLR is a top venue for cutting-edge research and technical innovation, the inclusion of novel technical designs would enhance the paper's impact and align with the high standards expected at such a conference.

- Q5: Could the authors provide detailed code and datasets to enable readers to reproduce the results? Based on the current information in the manuscript, it appears challenging for readers to replicate the experimental outcomes, which may hinder effective academic communication. Sharing reproducible resources would greatly enhance transparency and facilitate further research in this area.

---

> ### Comment · Reviewer_CK4W · 2024-11-21
> **Rebuttal?**
>
> I do not seem to have received any rebuttal from the authors.

---

### Official Review · Reviewer_WEG7 · 2024-11-04

**Soundness:** 3
**Presentation:** 3
**Contribution:** 2
**Rating:** 5
**Confidence:** 4

**Summary:**

This paper proposes a new benchmark for prompt injection attacks on LLMs.
Compared to existing works, the proposed benchmark is interactive and dynamic
and it has larger coverage on differerent modalities and tasks.

**Strengths:**

1. The studied problem is interesting.

2. The proposed benchmark in this paper has large coverage on differerent
modalities and diverse tasks.

**Weaknesses:**

1. The diversity of the models involved in the experiments could be improved.
Only three models are involved in the experiments, i.e., GPT-4o mini, LLaMA 3
and Qwen-VL, which may not provide sufficient coverage for a comprehensive
benchmark study. Expanding the experiment to include a broader range of models
is recommended. Additionally, specifying which particular models were used
within the LLaMA 3 and Qwen-VL families would improve clarity.

2. It is suggested to provide a more in-depth analysis of the benchmark results to
extract key observations and conclusions. For example, this could include
investigating the underlying causes of variations in attack success rates across
different modalities, models, and attack types.

3. The novelty of this paper might be incremental. A key contribution it highlights
is a dynamic framework with interactive environments. However, as recognized
in this paper, the existing work AgentDojo already provides similar
functionality. While this paper argues that AgentDojo lacks comprehensive
coverage of task modalities and unified analysis across different systems, these
limitations might be relatively minor given the technical contributions.
Expanding coverage to include more modalities and tasks may also be seen as an
incremental enhancement.

**Questions:**

please refer to Weaknesses

---

### Official Review · Reviewer_UV6S · 2024-11-04

**Soundness:** 3
**Presentation:** 3
**Contribution:** 2
**Rating:** 5
**Confidence:** 2

**Summary:**

This paper introduces FSPIB, a benchmark designed to evaluate foundational models when integrated with various modalities regarding prompt injection vulnerabilities. While prompt injection research has primarily focused on text-based models, this work explores how such attacks impact other modalities, such as documents, web pages, code, and images. The paper categorizes prompt injection threats and defenses into two levels: 1) Information level (e.g., information leakage, goal hijacking, response refusal) and 2) Action level (e.g., adversarial actions, parameter manipulation). Overall, the work seeks to provide a unified analysis of prompt injection vulnerabilities across diverse modalities, applications, and agents.

**Strengths:**

1. The paper advances current benchmarks by addressing vision, code, and web-based threats in addition to text-based modalities, and provides a structured analysis of both information-level and action-level threats.
2. The paper presents a comprehensive benchmark and analysis for prompt injection threats, covering a wide range of real-world applications, attacks, and defenses.
3. The interactive systems in a multi-turn evaluation add depth to the analysis, though specific details on the multi-turn evaluation are not fully elaborated.

**Weaknesses:**

1. This work reads more like a technical report exploring prompt injection across various applications and agents. The novelty needs to be clarified; for instance, what specific challenges arise in integrating various attacks into a unified benchmark? Further elaboration on the design rationales for the selection of agents, attacks, and defenses would be great.
2. The key takeaways from the experimental results are unclear. It would be great for the paper to provide more detailed and multi-dimensional discussions.
3. Prompt injection attacks may also be affected by the location of the prompts. (start, end, middle) and optimizing the adversarial prompt. Though some analysis is done, a thorough analysis would be great for prompt-injection benchmarks.

**Questions:**

1.	Please clarify the experimental settings in multi-turn evaluation.
2.	What are the major findings in the experimental results? Which applications or agents are more vulnerable?
3.	Is any specific integration with foundation models more vulnerable?

---

### Meta-Review · Area_Chair_RD4x · 2024-12-20

**Metareview:**

The paper introduces FSPIB, a benchmark for evaluating prompt injection vulnerabilities in FM-integrated systems across various modalities, including text, code, vision, and web-based applications. FSPIB categorizes threats into information-level and action-level types, offering a unified framework for assessing vulnerabilities, baseline defenses, and their effectiveness. It emphasizes dynamic and interactive testing environments, demonstrating the ongoing risks of prompt injection despite advancements in FM-based security.

Strength:
- The benchmark expands beyond text-based evaluations, covering diverse modalities such as code, web-based applications, and vision, providing a broader analysis.
- The structured categorization of information- and action-level threats provides a clear framework for analyzing prompt injection vulnerabilities.

Weakness:
- The contribution may be perceived as incremental due to overlap with existing tools like AgentDojo, with expanded modality coverage being the primary addition.
- The study involves only three models (GPT-4o mini, LLaMA 3, and Qwen-VL), limiting generalizability. Details about specific variants within these models are also missing.
- Key insights from experimental results are underexplored, with limited analysis of attack variability across modalities, tasks, and models.

**Additional Comments On Reviewer Discussion:**

No rebuttal provided by the authors.

---

### Decision · Program_Chairs · 2025-01-22

Reject